# Respiratory Syncytial Virus in Veneto Region: Analysis of Hospital Discharge Records from 2007 to 2021

**DOI:** 10.3390/ijerph20054565

**Published:** 2023-03-04

**Authors:** Silvia Cocchio, Gian Marco Prandi, Patrizia Furlan, Giovanni Venturato, Mario Saia, Thomas Marcon, Giulia Tremolada, Vincenzo Baldo

**Affiliations:** 1Department of Cardiac Thoracic Vascular Sciences and Public Health, University of Padua, 35131 Padua, Italy; 2Department for Woman and Child Health, University of Padua, 35128 Padua, Italy; 3“Azienda Zero” of Veneto Region, 35100 Padua, Italy

**Keywords:** respiratory syncytial virus, hospitalizations, Veneto region, trends, infants

## Abstract

Respiratory Syncytial Virus (RSV) is a known cause of acute lower respiratory infections in infants and young children. The present study aims to analyze the temporal trends and characteristics of hospitalization related to RSV in the Veneto region (Italy) in the period between 2007 and 2021. The analysis is performed on all the hospital discharge records (HDRs) of public and accredited private hospitals corresponding to hospitalizations occurring in the Veneto region (Italy). HDRs are considered if they included at least one of the following ICD9-CM codes: 079.6—Respiratory Syncytial Virus (RSV); 466.11—acute bronchiolitis due to RSV; and 480.1—pneumonia due to RSV. Total annual cases, sex, and age-specific rates and trends are evaluated. Overall, an increasing trend in the number of hospitalizations due to RSV was observed between 2007 and 2019, with a slight drop in RSV seasons 2013-2014 and 2014-2015. From March 2020 to September 2021, almost no hospitalization was registered, but in the last quarter of 2021, the number of hospitalizations reached its highest value in the series. Our data confirm the preponderance of RSV hospitalizations in infants and young children, the seasonality of RSV hospitalizations, and acute bronchiolitis as the most frequent diagnosis. Interestingly, the data also show the existence of a significant burden of disease and a non-negligible number of deaths also in older adults. The present study confirms RSV is associated with high rates of hospitalization in infants and sheds light on the burden in the 70+ age group in which a considerable number of deaths was observed, as well as the parallelism with other countries, which is consistent with a wide underdiagnoses issue.

## 1. Introduction

Respiratory Syncytial Virus (RSV) is a major cause of acute lower respiratory infections (ALRIs) especially in infants and young children, with more than 60% and 80% of all children estimated to be infected, respectively, within their first and second year of age [1,2]. The virus is also related to lower respiratory tract morbidity in elderly individuals [3]. The infection is known to be associated with mild symptoms, but it constitutes the most frequent cause of hospitalization due to ALRI in children under 2 years of age [2,4]. The main risk factors for severe forms of the disease in children are prematurity, bronchopulmonary dysplasia, and congenital heart disease; other risk factors for RSV infection and consequent complications include low birth weight, living with older siblings, exposure to the smoke of cigarettes, and a lack of breastfeeding [3,5]. In adults, higher rates of disease are observed in patients affected by chronic pulmonary diseases, chronic cardiac diseases, or immunodeficiency [3]. In pre-SARS-CoV-2 pandemic years, RSV accounted for an estimated 33.1 million cases worldwide per year in children below the age of 5 years [6], while the infection’s frequency in older adults, even if not completely known, was estimated, in industrialized countries alone, at 1.5 million cases per year [7]. The methodological differences across studies and variations in surveillance protocols might render the actual incidence difficult to gauge. Currently, no up-to-date general recommendations on RSV surveillance are available for member states of the European Union [8]. In Italy, an RSV surveillance system has not been implemented at the national level, and data about the spread and impact of the pathogen, where available, are collected through the surveillance systems for flu and influenza-like illness (ILI). Data collected through the surveillance systems for flu and ILI indicate that aerial viruses, such as RSV, typically circulate from late October to early March, the year-time commonly defined as “RSV season”, and infection numbers reach peaks in December and January [9].

In Italy, hospital discharge records (HDRs) routinely collect administrative and clinical information on all hospital stays in public and private hospitals throughout the country, including procedures and operations coded in accordance with the ICD-9-CM international standard [10]. Despite the fact that HDRs are not systematically used to assess RSV hospitalization burden, they can be a useful tool for retrospective surveys on severe cases of ALRI, especially for children and infants [11,12].

Accordingly, the present study aims to analyze temporal trends and characteristics of ALRI hospitalizations related to RSV in the Veneto region (Italy) between January the 1st 2007 and December the 31st 2021 through HDR data. In addition, an assessment of the possible role of the COVID-19 epidemic in modifying RSV-related hospitalization rates (HRs) is carried out.

## 2. Materials and Methods

We conducted the analysis on data from January the 1st 2007 to December the 31st 2021 in the Veneto region. HDRs of Veneto’s database were checked to include codes of the International Classification of Diseases, Ninth Revision, Clinical Modification (ICD-9-CM) linked to RSV-related diseases. Hospitalizations were included in the analysis if the corresponding HDRs had at least one of the following ICD9-CM codes: 079.6—Respiratory Syncytial Virus; 466.11—acute bronchiolitis due to RSV; and 480.1—pneumonia due to RSV. We considered two hospitalizations due to RSV for the same subject within a 30-day period as repeated admissions, and in those events, we only included in the analysis data from the first hospital stay. We also excluded from the analysis hospital admissions related to day-care treatments, i.e., those treatments that did not require hospitalization.

Through HDRs, we collected demographic data (date of birth, gender, and place of residence) and clinical information (clinical manifestation and discharge or death). Annual hospitalization rates were estimated by dividing the number of hospitalizations in a given season by the population of Veneto residents in the designated year as per data from the Veneto Regional Authority of Statistics office [13]. We conducted analysis by using seven age groups according to the age at the time of the hospital (<1 year; 1–4 years; 5–49 years; 50–69 years; and older than 70 years). Such a stratification allowed the direct analysis of trends of RSV diseases in age groups.

Data were analyzed by using Student’s t-test for continuous data and Pearson’s chi-square test for categorical data. To evaluate the correlation between the length of stay and age, linear regression was performed, obtaining a Beta (B) coefficient. A *p*-value < 0.05 was considered significant. The analyses were performed using the Statistical Package for the Social Sciences (SPSS 28.0; SPSS Inc., Chicago, IL, USA). Significant trends over the years considered were assessed as average annual percent changes (AAPCs), which is a summary measure of the trend over a given fixed interval that is computed as a weighted average of the annual percent change (APC) emerging from the joint-point model, using weights equating to the length of the APC interval. If an AAPC lies entirely with a single joint-point segment, the AAPC is the same as the APC for that segment [14].

## 3. Results

From 1 January 2007 to 31 December 2021, 7581 hospitalizations whose HDRs included codes related to an RSV diagnosis in one of the primary or secondary fields were registered in the Veneto region. A total of 8.2% of all entries were excluded from the analysis, which eventually included data from 6961 hospitalizations, corresponding to a mean of 464 hospitalizations/year. Of the excluded HDRs, n. 323 were repeated admissions, a total of 96.6% (n. 312) were regarded as children <1 year of age, and a total of 88.5% (n.286) had a diagnosis of acute bronchiolitis (Figure 1).

The most frequent ICD-9 code source of entry was 466.11 (acute bronchiolitis due to RSV), which constituted 83.6% of the total with 5818 hospitalizations. Code 480.1 (pneumonia due to RSV) represented an additional 10.6% (n.741) and code 079.6 (Respiratory Syncytial Virus) the remaining 5.8% (n.402). With respect to the field of diagnosis, overall, a total of 65.5% of HDR was in primary diagnosis, specifically, 68.2%, 70.7%, and 16.9% for acute bronchiolitis and pneumonia due to RSV and RVS. In Figure 2, the distribution of RSV-related HDRs for age groups is shown.

Table 1 shows the characteristics of RSV-related hospitalizations by diagnosis. In terms of gender, we observed minor differences: 54.2% (n.3774) of hospitalizations occurred in males and the remaining 45.8% (n.3187) in females. In terms of age, we observed a clear preponderance of cases in children versus adults, and the biggest proportion, 83.9% (n.5840), was registered in infants. Children aged 1 to 4 years represented an additional 10.9% (n.756) of the total. The proportion of hospitalizations in older subjects was almost negligible: 1.7% (n.115) occurred in subjects aged 5–49 years, 1.2% (n.81) in those aged 50–69 years, and the remaining 2.4% (n.169) in the ones aged 70 years or more. Referring to the diagnosis of admission, the frequency of acute bronchiolitis is significantly higher for infants than the other two diagnoses: 93.0% vs. 38.7% for pneumonia, *p* < 0.001, and 93.0% vs. 35.1% for RSV, *p* < 0.001. On the other hand, the age groups of 1-4 years and 70+ years were admitted with a percentage significantly higher for pneumonia and RSV than acute bronchiolitis, *p* < 0.001. Altogether, a total of 95.6% of hospitalizations were recorded in the pediatric ward. Overall, the mean length of stay was 6.0 days; specifically, the hospitalizations due to pneumonia were significantly longer than both acute bronchiolitis and RSV hospitalizations, 8.0 vs. 5.7 *p* < 0.001 and 8.0 vs. 6.8, *p* = 0.011, respectively. Overall, the length increased with age (β (95%CI): 0.091 (0.081–0.101), *p* < 0.001); specifically, (β (95%CI): 0.080 (0.059–0.100), *p* < 0.001) for acute bronchiolitis, (β (95%CI): 0.109 (0.083–0.135), *p* < 0.001) for pneumonia, and (β (95%CI): 0.063 (0.045–0.081), *p* < 0.001) for RSV.

The hospitalizations of RSV observed in the Veneto region between 2007 and 2021 showed a well-defined seasonal pattern: infections are low between April and September, progressively rise and peak between December and January, and from February drop again (RSV season begins in October and runs through March of the next year). Such a pattern found only one interruption that occurred between April 2020 and September 2021: in 17 consecutive months, almost no hospitalization was registered. In the following quarters, i.e., the third and fourth quarters of 2021, 26 and 884 hospitalizations were registered, respectively, with the latter being the record high for the entire period. The main hospitalization group was consistently registered as the <1 year age group, and those within the 1–4 years age group constituted almost all the remaining portion. In other age groups, the number of hospitalizations was always in the scale of units, with two exceptions: the group of subjects aged 70 years and older and the group of subjects from 50 to 69 years. In the group 70+ years, 10 or more hospitalizations were registered from season 2016–2017 to season 2019–2020 and in season 2021–2022. In the group 50–69 years, 10 or more hospitalizations were registered in seasons 2017–2018, 2018–2019, and 2021–2022. Considering the fall–winter seasonality of the disease (the fourth and first quarters altogether) and looking at the hospitalization data by season, the number of hospitalizations showed a continuously rising trend from the 2007–2008 RSV season (n.203) to the 2019–2020 RSV season (n.761). In the RSV season 2020–2021, then, a strong drop was observed with an almost total zeroing in the number of hospitalizations (n.10). The partial data of 2021–2022 RSV season, however, indicate a new rise in events, with hospitalizations reaching the highest number of the entire series (n.884) (Figure 3).

Considering only 0–4 years of age, the proportion of the 1–4 age group from 2007 to 2018 was quite stable; on average, 9.8%, and from 2019 (11.7%), it increased, reaching 13.7% in 2020 and 19.8% in 2021 (APC (CI95%): 37.4 (−0.7; 90.2)). Focusing on the two-year period between January 2020 and December 2021, we observed a peculiar pattern in hospitalizations. From January to March 2020, the progressive drop in the number of hospitalizations is similar to that of the previous season but steeper. In April 2020, almost one month after the beginning of the lockdown (24), which included school, nursery, and kindergarten closures, only 2 hospitalizations occurred versus the 17 in April 2019, and in May 2020, just 1 was registered versus the 7 in May 2019. It must be noted that these three cases were registered exclusively in subjects 50 years of age or older. In the following four months, no hospitalization occurred. Between October 2020 and December 2020, only four hospitalizations occurred, and all of them were subjects 50 years of age or older. Of the 11 hospitalizations that occurred between January 2021 and July 2021, 5 were in children from 0 to 5 years and the remaining in adults 50 years or older. In August 2021, six hospitalizations occurred, and they were all in infants. Finally, from September 2021, the numbers started to rise again, with 18 cases in September 2021, 190 in October, 543 in November, and 151 in December.

Overall, the general population trend was positive through 2007 to 2019 (APC (CI95%): 9.8 (6.8; 12.9)). In terms of hospitalization rates for a single age group, the <1 year age group registered a hospitalization rate of 3.6/1000 in season 2007–2008 and one of 19.8/1000 in season 2021–2022. Between these two values, the hospitalization rate of a season was always lower than that of the next season except for season 2020–2021, when the rate was almost zero, and in seasons 2013–2014 and 2014–2015, when we observed a momentaneous reduction in hospitalization rates. In age group <1, the rate showed a significative positive trend (AAPC (CI95%): 13.4 (10.7; 16.2)). In the age group 1–4 years, the pattern is less defined, without relevant fluctuations until season 2013-2014, and from season 2014–2015 to season 2019–2020, the rates show a positive trend (APC (CI95%): 34.6 (12.7; 60.8)), a drop in season 2020–2021, and then reach their series record of 1.16/1000 in season 2021–2022. In other age groups, hospitalizations rates were negligible (i.e., <0.04/1000) for most of the period except for punctual registrations, such as a rate of 0.07/1,000 in season 2021–2022 recorded in the 5–9 years age group. A relevant finding is the registration of a significative positive trend in adults aged 70 years and more between 2014 and 2019 (APC (CI95%): 59.0 (38.3; 82.8)) (Figure 4).

In the period of study, we registered 23 deaths in patients hospitalized for an RSV-related ICD-9 code; this value corresponds to 0.3% of all hospitalizations (Table 1). The diagnosis was indicated as primary in five subjects (21.7%): one for acute bronchiolitis and four for pneumonia due to RSV, and all in subjects aged 70 years or more. Regarding age group, 5 (21.7%) occurred in infants (4 due to acute bronchiolitis and 1 to pneumonia due to RSV), 1 (4.3%) for pneumonia due to RSV in subjects aged 5 to 49 years, 4 (17.4%) in subjects aged 50 to 69 years (3 for pneumonia due to RSV and 1 to RSV), and 13 (56.5%) in subjects aged 70 years or more. The chronological distribution of deaths indicates that no death occurred in infants hospitalized for RSV-related codes after the 2014–2015 RSV season and that the eight deaths registered from RSV season 2016–2017 occurred exclusively in adults. Other than that, and considering the small number of events registered, no general trend was recognizable among seasons or subgroups. Eight deaths were related to hospitalization for acute bronchiolitis: four of them occurred in infants and four in people aged 70 years or more. A total of 14 deaths were related to a pneumonia diagnosis: 1 occurred in the <1 year age group, 1 in the 05–49 years age group, 3 in the 50–69 years age group, and 9 in the 70 years and older age group. Only one death, registered in the 50–69 years age group, was associated with hospitalization for the code 079.6 (Respiratory Syncytial Virus). Of the 23 deaths, n.5 (21.7%) were recorded in the primary field, specifically 1 for acute bronchiolitis and 4 for pneumonia (Figure 5).

## 4. Discussion

In the period from January 2007 to December 2021, we observed an overall increase in the number of hospitalizations for RSV-related diseases in the Veneto region, with the exception of RSV season 2020–2021. We observed an increase from 225 hospitalizations counted in RSV season 2007–2008 to 884 in RSV season 2021–2022, which is, however, partial datum, as it consists only of events registered in the last quarter of 2021. The increase in the number of overall hospitalizations is almost completely due to the increase in numbers in the pediatric age group and, in particular, in infants.

The RSV-related ICD-9 codes registered in the Veneto region in the period of analysis show a predominance of hospitalizations associated with bronchiolitis in infants. In fact, this entry alone represented 83.6% of all hospitalization records. Such data are in line with the known characteristics of RSV infection and its clinical manifestations that are considered a peculiarity of infants and, for this reason, largely investigated, probably only in this part of the population in industrialized countries [15,16,17]. Diagnostic criteria are usually defined by clinical judgment, and our data did not allow linkage with laboratory testing, which is not routinely recommended. Therefore, it is reasonable to assume that RSV infections in older in-patients might be underestimated, although it is difficult to gauge by how much. Our analysis shows an increasing trend in RSV-related hospitalizations in the 70+ years age group for RSV season 2014–2015. The increase may be related to numerous factors, one of which is probably the effect of the improving sensitivity of the diagnostic process in adults and older adults with ALRI.

Considering the variable primary or secondary diagnosis, no clear patterns emerge if the finding that acute bronchiolitis and pneumonia are consistently reported as a primary diagnosis in age groups <1 year and 1 to 4 years and the finding that RSV is more frequently listed as a secondary diagnosis in all other age groups. Such a finding can be explained partly by the epidemiology of the virus itself, and partly by the diminished consideration RSV has as a cause for disease in older children, adults, and older adults.

The rates of hospitalization in the age groups <1 year and 1–4 years are consistent with those registered for the same seasons in a previous analysis of RSV-related ICD-9 codes in the Veneto region performed from RSV season 2012–2013 to RSV season 2016-2017 [12]. Both that analysis and ours find similar rates of hospitalization in infants, and both highlight a slight concavity of the curve in RSV season 2013–2014 and 2014–2015 followed by a rise. More specifically, in the age group <1 year, we estimated a decrease in hospitalization rates from 8.4/1000 population in RSV season 2012–2013 to 7.3/1000 population in RSV season 2013–2014 to 6.4/1000 population in RSV season 2014–2015. The cause of said concavity might be complex to assess, while in the literature, changes in RSV seasonal trends have been related to a variety of factors, from temperatures to pollution. Nonetheless, our extended period of analysis confirms a pattern of increased rates of hospitalization in infants with the exception of RSV season 2020–2021. The drop in hospitalization observed between March 2020 and July 2021 may be explained mainly by considering the limitations to social activities and human interactions that occurred in the period. In fact, while full lockdown measures were implemented only from March 2020 to May 2020, mandatory face-mask utilization was protracted through 2020 and 2021, and other measures, such as school closures and class disruptions, were occasionally reinforced through 2020 and 2021 in case of local clusters of infection. All in all, these limitations to inter-personal interactions may explain the limited circulation of the virus and the consequent presence, in the second half of 2021, of a wide part of the population naïve to the virus and, therefore, more susceptible. Such a hypothesis may be useful in interpreting the modification in the proportion of hospitalizations for the 1 to 4 years age group. Considering hospitalizations only in subjects aged 0 to 4 years, in fact, the proportion of those that occurred in the 1–4 age group remained stable from 2007 to 2018, being on average 9.8%. From 2019, there was a first hint of increasing percentages (11.7%), and the proportion continue to rise in 2020 (13.7%) and reached a value of 19.8% in 2021 (APC (CI95%): 37.4 (−0.7; 90.2)). The positive trend in hospitalization rates in infants seems to be observed only in Italy. In fact, an analysis of data from six other European countries shows biennial patterns in RSV-coded hospitalization rates in Finland, Norway, and Denmark and flat trends in England, Scotland, and the Netherlands [11]. A possible explanation for the increasing trend has been advanced in an Italian study that retrospectively evaluated a population of Italian children aged 0–6 years with a laboratory-confirmed diagnosis of RSV infection and who were hospitalized between September 2014 and August 2019. As the authors observed a progressive increase in the number of positive cases in the study period, they formulated the hypothesis of a parallel progressive increase in the number of tests requested by clinicians over the course of the study period as well as of improvements in laboratory techniques [1]. This hypothesis resonates with numerous, yet anecdotal, observations reported by pediatricians in Veneto.

Hospitalizations in older children and young adults constitute a very small fraction of the total, and their numbers are consistent with the results of the abovementioned Italian study, which registered only sporadic cases in children aged between 3 and 6 years [1]. On the other hand, hospitalization rates in older adults show the existence of a burden of disease also in this segment of the population. This observation is only partially aligned with the results of a retrospective review of hospitalizations registered in Australia between 2006 and 2015 in subjects aged 65 years and older. The mean Australian hospitalization rate, in fact, resulted in 21/100,000 population [18] versus the Veneto rate that ranges between 0.1/100,000 and 3.8/100,000. Moreover, our results are markedly lower than those of a model performed on the United Kingdom population, which estimated, in the period between 1995 and 2009, a hospitalization rate of 156/100,000 population for adults aged 65 or more [19], and even lower with respect to a hospitalization rate of 254/100,000 population reported in a study carried out in the USA [20]. Nevertheless, the increase in records of RSV-related hospitalizations in Veneto registered from RSV season 2014–2015 to RSV season 2019-2020 as well as the 33 cases registered in the fourth quarter of 2021 must be noted. While the increased number of entries may be interpreted, similarly to what happened in Australia, as an improved recognition of RSV and testing for the related disease also in older adults [21], the general and historically low awareness of the disease in doctors treating older adults may explain the discrepancy between numbers of hospitalization reported in Italy and the numbers reported in other countries.

In terms of RSV-associated deaths, we observed some peculiarities. First, in the period between 2007 and 2017, 13 deaths were registered, and 8 of them were associated with hospitalizations for bronchiolitis and 5 with pneumonia. All these deaths, except one, occurred in infants (<1 year of age) and in older adults (>70 years of age). Second, in the period between 2019 and 2021, 10 deaths occurred, and 9 of them were associated with pneumonia and 1 with a nonspecific diagnosis of RSV. Interestingly, none of the deaths registered between 2019 and 2021 occurred in infants, only one in subjects aged 5 to 49 years of age, three in subjects 50 to 69 years of age, and six in subjects aged 70 years or more. Considering the number of deaths and the number of hospitalizations, we observed a dramatic disproportion in the different age groups; in fact, the 5.840 hospitalizations in infants were associated with 5 deaths with a ratio of less than 1 death every 1.000 hospitalizations. In the 70+ years age group, instead, 169 hospitalizations were registered deaths, with a ratio of 1 death every 15.4 hospitalizations. Similarly, in adults aged 50 to 69 years, 4 deaths were registered in the 81 hospitalizations reported, with a rate of 1 death every 20.2 hospitalizations. Even if these numbers and ratios are not expressive of statistical differences between age groups, they can hint at some possible and non-mutually exclusive hypotheses, such as a different degree of fatality of the RSV infection in older patients, the late hospital admission of older patients affected by RSV-diseases and, therefore, the more frequent occurrence of severe cases in older patients, and the underdiagnosis of RSV in this part of the population. The latter hypothesis seems to be confirmed by the previously mentioned analysis of Saravanos et al. [18], Fleming et al. [19], and Widmer et al. [20], while a high level of fatality rates in adults and older adults were observed also in Spain between 2012 and 2020 [22].

The main strength of the present analysis is the inclusion of data covering fifteen years, fourteen complete RSV seasons, and the first part of RSV season 2021–2022. The results of the analysis shed light on the burden of disease in the different age groups but leave open the question about the representativeness of those routinely collected data on the epidemiology of RSV. In fact, on the one hand, by using ICD-9 codes, we profit from a commonly used and well-implemented source of data that enables us to define trends and patterns over time. On the other hand, many factors can affect the data during their generation; among them, the most important one is the lack of the application of testing in older children and adults. Nevertheless, this study highlights the same patterns of seasonality and age distribution of those found by other studies within the same region or similar countries. Moreover, our results confirm the high burden of RSV in children aged <1 year and the peak of the disease in the first 3 months of infancy.

## 5. Conclusions

This study highlights the relevance of RSV in terms of hospitalizations in the general population in the Veneto region. Across the 15 years of data in analysis, the preponderance of cases in the pediatric population can be confirmed, and this preponderance follows a clear positive trend through the years, which is probably related more to improved diagnostic procedures than to an actually increased burden of RSV infections. From 2014, an increasing frequency of hospitalizations was also observed in adults, especially in the 70+ years age group, in which we registered the major parts of deaths in RSV hospitalizations. In the older segment of the population, however, the discrepancy between a relatively high number of deaths and a small number of hospitalizations seems to indicate a significant underdiagnosis of diseases due to RSV in this age group. A clearer picture of the burden of the disease may come from analyses specifically designed to investigate the older segment of the population as well as from the implementation of mathematical models.

## Figures and Tables

**Figure 1 ijerph-20-04565-f001:**
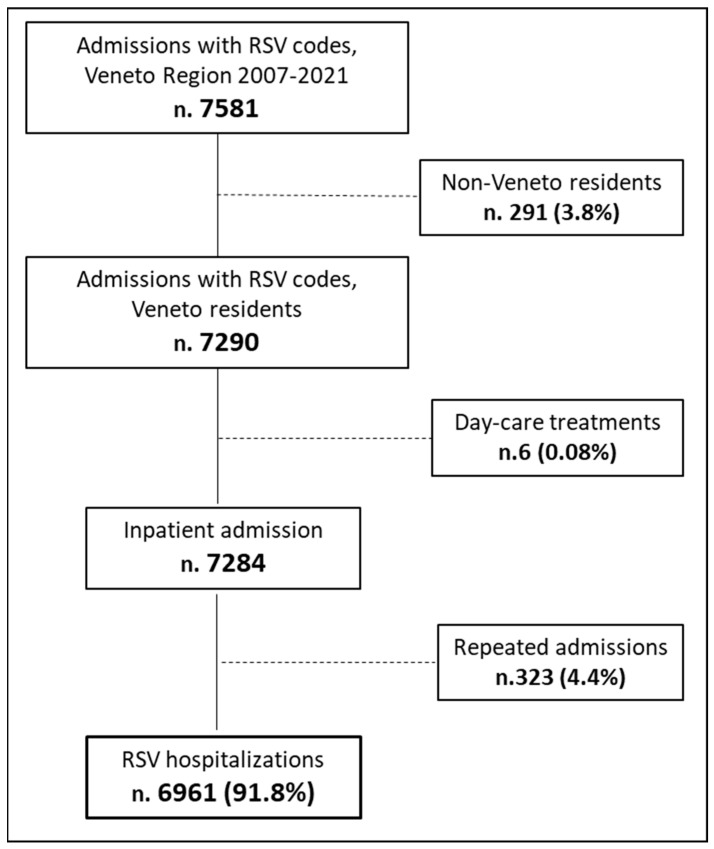
Flow chart of the study population selection.

**Figure 2 ijerph-20-04565-f002:**
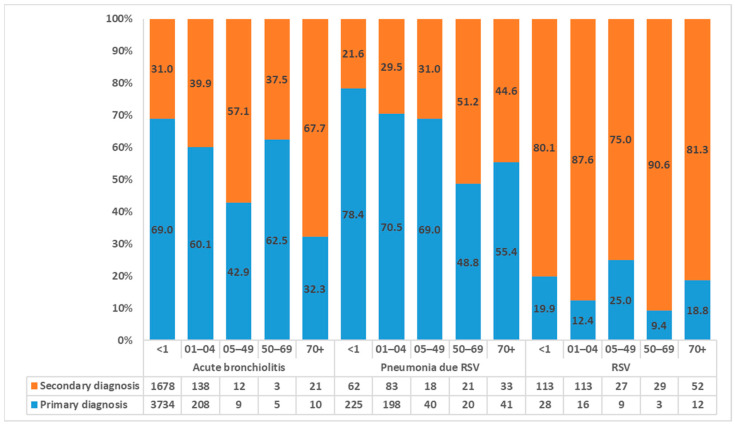
Distribution of RSV-related hospitalizations by diagnosis, age group, and field of diagnosis.

**Figure 3 ijerph-20-04565-f003:**
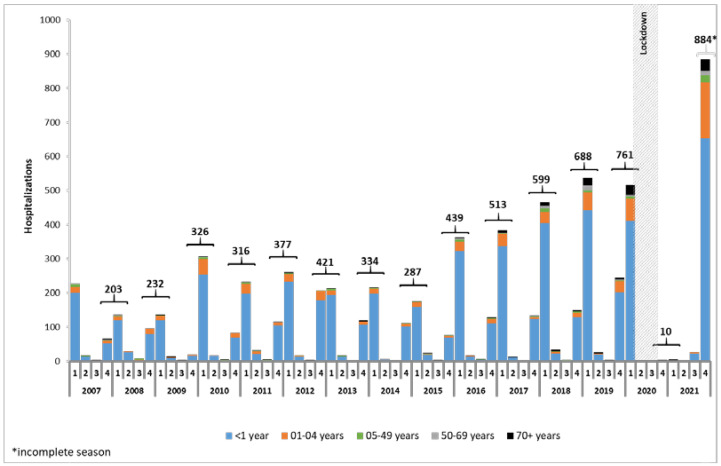
Distribution of RSV-related hospitalizations between 2007 and 2021 by quarter and for age group.

**Figure 4 ijerph-20-04565-f004:**
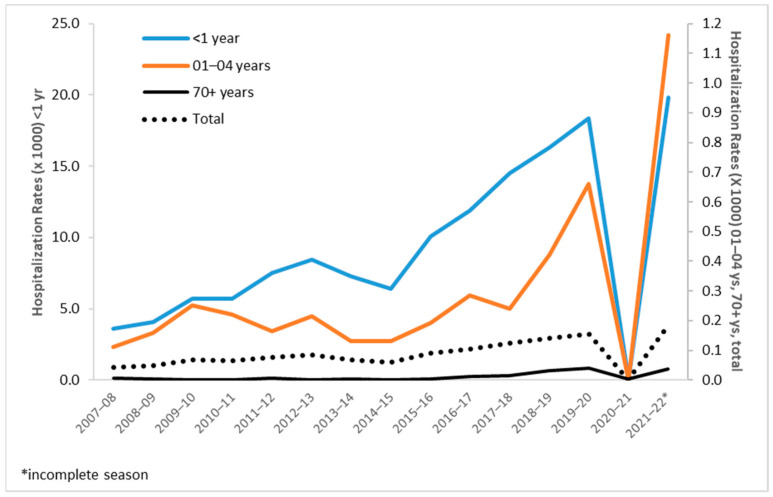
Hospitalization rates by RSV-season and age group.

**Figure 5 ijerph-20-04565-f005:**
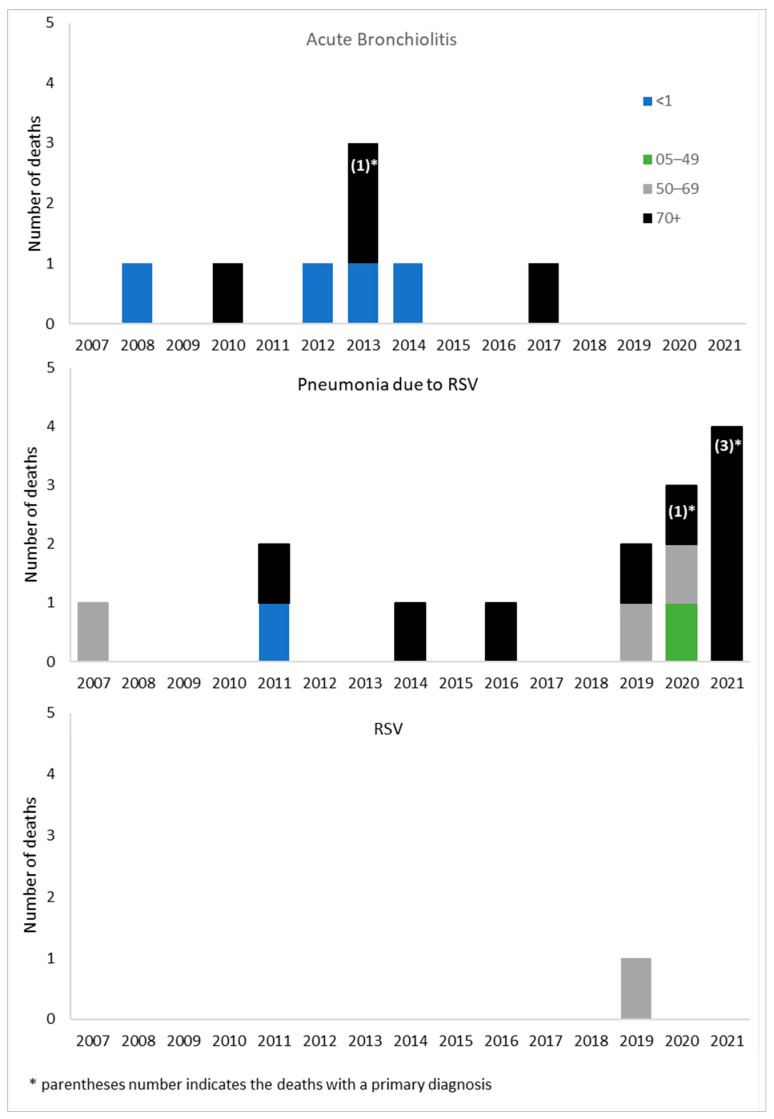
Distribution of death by year, age group and diagnosis.

**Table 1 ijerph-20-04565-t001:** Characteristics of the hospitalizations considered, by diagnosis of admission.

Variables	Acute Bronchiolitis (n. 5818)		Pneumonia (n. 741)		RSV (n. 402)		Total (n. 6961)	
	N	(%)	n	(%)	n	(%)	n	(%)
Gender								
Males	3205	(55.1)	366	(49.4)	203	(50.5)	3774	(54.2)
Females	2613	(44.9)	375	(50.6)	199	(49.5)	3187	(45.8)
Age group								
<1 year	5412	(93.0)	287	(38.7)	141	(35.1)	5840	(83.9)
01–04 years	346	(5.9)	281	(37.9)	129	(32.1)	756	(10.9)
05–49 years	21	(0.4)	58	(7.8)	36	(9.0)	115	(1.7)
50–69 years	8	(0.1)	41	(5.5)	32	(8.0)	81	(1.2)
70+ years	31	(0.5)	74	(10.0)	64	(15.9)	169	(2.4)
Quarter								
First	3547	(61.0)	388	(52.4)	241	(60.0)	4176	(60.0)
Second	208	(3.6)	40	(5.4)	18	(4.5)	266	(3.8)
Third	58	(1.0)	13	(1.8)	7	(1.7)	78	(1.1)
Fourth	2005	(34.5)	300	(40.5)	136	(33.8)	2441	(35.1)
Admission wards								
Pediatrics	5472	(94.1)	569	(76.8)	271	(67.4)	6312	(90.7)
Intensive care	160	(2.8)	26	(3.5)	8	(2.0)	194	(2.8)
Neonatal intensive care	85	(1.5)	6	(0.8)	6	(1.5)	97	(1.4)
Neonatology	46	(0.8)	2	(0.3)	1	(0.2)	49	(0.7)
General medicine	21	(0.4)	75	(10.1)	72	(17.9)	168	(2.4)
Pneumology	14	(0.2)	12	(1.6)	16	(4.0)	42	(0.6)
Geriatric care	7	(0.1)	10	(1.3)	8	(2.0)	25	(0.4)
Infectious disease	1	(0.0)	12	(1.6)	14	(3.5)	27	(0.4)
Other	12	(0.2)	29	(3.9)	6	(1.5)	47	(0.7)
Mean length of stay	5.7		8.0		6.8		6.0	
Death	8	(0.1)	14	(1.9)	1	(0.2)	23	(0.3)

## Data Availability

The data supporting the findings of this study are available from the corresponding author upon reasonable request, and first has to be approved by Azienda Zero (Veneto region).

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
