# Peer review of "Respiratory Syncytial Virus in Veneto Region: Analysis of Hospital Discharge Records from 2007 to 2021"

_ijerph, 2023, doi:10.3390/ijerph20054565_

Round 1
Reviewer 1 Report
This paper presents results from analysis of hospital discharge records for RSV-related admissions in Veneto, Italy from 2007 to 2021. Overall, the paper represents an important source of data, particularly regarding the overlooked burden of RSV in older adults. The paper would benefit from some additional sensitivity analyses and additional discussion of the impact of the COVID-19 pandemic. The paper should also be carefully proofread. The paper should be considered for publication after these revisions. I have provided some specific comments and suggestions below.
Introduction
Line 53: It would be helpful to provide some context about the RSV season in Italy. Such as “In Italy, RSV typically circulates from November – March, with a peak in January. There is no routine surveillance system at the national-level and data about the spread….”
Lines 65 – 67: This is a very important goal of the paper, but apart from mentioning the low number of hospitalizations in 2020-21 and the large rebound in Q4 2021, there is no real discussion about the role of the COVID-19 pandemic in driving this. I appreciate that the lockdown period was included in Figure 3, but more narrative description of the timing of the lockdown and why circulation remained low even after the conclusion of lockdown (perhaps due to other non-pharmaceutical interventions?), would provide richness and context to the discussion.
Methods
Lines 73-74: Perhaps as a sensitivity analysis the authors could consider comparing the results for the RSV codes to results when all bronchiolitis and all pneumonia records are considered. Are there trends in these broader categories? If not, that would support the claim that much of the upward trend is due to changes in testing and reporting practices vs. actual changes in viral circulation (which I agree is the most likely explanation in the years prior to the SARS-CoV-2 pandemic).
Line 79: Please provide more explanation regarding the definition of a “day-care treatment”.
Lines 84 – 87: The authors could consider combining some of the middle age groups. This could help to simplify the results and overall message, serving to further emphasize the burden in young children and older adults.
Lines 91-96: I don’t see the results from this part of the analysis in the results section. I only see the mention of a rising trend (line 154), but nothing about AACP. Please include in the results.
Results
Line 99: Authors should consider a sensitivity analysis stratifying based on if the RSV code was listed in the primary or secondary field. At the very least they should provide a breakdown by age groups for the proportion of HDRs where RSV was listed in the primary vs. secondary field.
Line 113: The authors could combine the middle age groups to make a 5-49 age group (as they present in the results here) and the methods should be revised accordingly, as suggested above.
Lines 120-122: Might the length of stay be confounded by age? If you stratify by age or adjust for age is the pattern the same?
Lines 126 – 136: The authors provide a very detailed breakdown of these 23 deaths, one additional thing to include would be the proportion where RSV is listed in primary vs secondary field.
Line 135: I would be hesitant to give this level of detail (exact age) of a single patient.
Line 139: Is it really two weeks or a single peak that extends from the fourth quarter into the first quarter of the following year. I suggest the authors define the RSV season in Italy in the introduction or methods.
Line 141: Instead of saying “minimal” perhaps provide a threshold such as “always less than x”
Lines 145 -147: Please note if the proportion of RSV hospitalizations in those 1-4 changed in the 2021 rebound. There has been a lot of discussion around the impact of the missed 2020-21 RSV season and the delay age at first infection. Many expected a rise in hospitalizations in this older age group, does your data support this?
Discussion – general: While the authors note the drop in RSV in the 2020-2021 season and the rebound in 2021 Q4, I don’t see any mention of the SARS-CoV-2 pandemic and the role of non-pharmaceutical interventions in depressing the circulation of endemic respiratory pathogens or disruptions in seasonality (2021 appears to have a November peak compared to January peaks in previous years). This pattern has been seen in many places and should be added to the discussion.
Lines 222 – 224: This would be a good place to include any changes you saw in the 1-4 age group during Q4 2021 – after the missed 2020-21 RSV season, thus delaying age of first infection/hospitalization.
Line 230 – 238: I agree that improved recognition of the burden of RSV in older adults probably explains the increasing trend. But any thoughts on why your estimates for this age group are so much lower than the other studies?
Lines 241 – 244: The authors do not mention the shift in cause of death from bronchiolitis to pneumonia after 2017 (which seems to also coincide with a shift in age). Would be worth mentioning in the results if it is going to be a focus in the discussion.
Lines 246 – 251: These sentences need revision so that punctuation is consistent. The number of deaths in 70+ is missing. It would also be more appropriate to give these detailed level figures in the results and focus on patterns in the discussion.
Lines 251-256: I think this is the most important part of the paper. RSV is only recently being acknowledged as a serious threat in older adults. Highlighting this is extremely important, especially since vaccines for seniors will be available soon.
Author Response
Reviewer #1
This paper presents results from analysis of hospital discharge records for RSV-related admissions in Veneto, Italy from 2007 to 2021. Overall, the paper represents an important source of data, particularly regarding the overlooked burden of RSV in older adults. The paper would benefit from some additional sensitivity analyses and additional discussion of the impact of the COVID-19 pandemic. The paper should also be carefully proofread. The paper should be considered for publication after these revisions. I have provided some specific comments and suggestions below.
Introduction
Line 53: It would be helpful to provide some context about the RSV season in Italy. Such as “In Italy, RSV typically circulates from November – March, with a peak in January. There is no routine surveillance system at the national-level and data about the spread….”
Authors’ response: Thanks for the comment. The introduction has revised by adding the relevant information, lines 56-58.
Lines 65 – 67: This is a very important goal of the paper, but apart from mentioning the low number of hospitalizations in 2020-21 and the large rebound in Q4 2021, there is no real discussion about the role of the COVID-19 pandemic in driving this. I appreciate that the lockdown period was included in Figure 3, but more narrative description of the timing of the lockdown and why circulation remained low even after the conclusion of lockdown (perhaps due to other non-pharmaceutical interventions?), would provide richness and context to the discussion.
Authors’ response: Thanks for the comment. We added lines 362-369.
Methods
Lines 73-74: Perhaps as a sensitivity analysis the authors could consider comparing the results for the RSV codes to results when all bronchiolitis and all pneumonia records are considered. Are there trends in these broader categories? If not, that would support the claim that much of the upward trend is due to changes in testing and reporting practices vs. actual changes in viral circulation (which I agree is the most likely explanation in the years prior to the SARS-CoV-2 pandemic).
Authors’ response: As suggested we compared the RSV trends with all bronchiolitis and all pneumonia records but for the latter the AAPC was stable over period (AAPC:1.4 (-2.2;5.0). The discussion was revised
Line 79: Please provide more explanation regarding the definition of a “day-care treatment”.
Authors’ response: As suggested we added the information (line 82)
Lines 84 – 87: The authors could consider combining some of the middle age groups. This could help to simplify the results and overall message, serving to further emphasize the burden in young children and older adults.
Authors’ response: Thanks for the suggest, we combined the middle age group in 05-49 years, Table 2; Figure 4 and Figure 5 and in the text, results and methods
Lines 91-96: I don’t see the results from this part of the analysis in the results section. I only see the mention of a rising trend (line 154), but nothing about AACP. Please include in the results.
Authors’ response: We included AAPC in the results
Results
Line 99: Authors should consider a sensitivity analysis stratifying based on if the RSV code was listed in the primary or secondary field. At the very least they should provide a breakdown by age groups for the proportion of HDRs where RSV was listed in the primary vs. secondary field.
Authors’ response: Thanks for suggest, we added information in Figure 2 (new) and in results section
Line 113: The authors could combine the middle age groups to make a 5-49 age group (as they present in the results here) and the methods should be revised accordingly, as suggested above.
Authors’ response: Thanks for the suggest, we combined the middle age group in 05-49 years, Table 2; Figure 4 and Figure 5 and in the text, results and methods
Lines 120-122: Might the length of stay be confounded by age? If you stratify by age or adjust for age is the pattern the same?
Authors’ response: Thanks for suggest, we made linear regression to evaluate correlation between length of stay and age. We’ve added in method and in results section
Lines 126 – 136: The authors provide a very detailed breakdown of these 23 deaths, one additional thing to include would be the proportion where RSV is listed in primary vs secondary field.
Authors’ response: Thanks for the comment. We added in results section and in the new Figure 3
Line 135: I would be hesitant to give this level of detail (exact age) of a single patient.
Authors’ response: Thanks for the comment. It has been removed from the manuscript.
Line 139: Is it really two weeks or a single peak that extends from the fourth quarter into the first quarter of the following year. I suggest the authors define the RSV season in Italy in the introduction or methods.
Authors’ response: Thanks for the suggest. We corrected lines 56-58 and lines 206-208
Line 141: Instead of saying “minimal” perhaps provide a threshold such as “always less than x”
Authors’ response: Thanks for the comment. We have changed it.
Lines 145 -147: Please note if the proportion of RSV hospitalizations in those 1-4 changed in the 2021 rebound. There has been a lot of discussion around the impact of the missed 2020-21 RSV season and the delay age at first infection. Many expected a rise in hospitalizations in this older age group, does your data support this?
Authors’ response: Thanks for the comment. We added lines 253-255
Discussion – general: While the authors note the drop in RSV in the 2020-2021 season and the rebound in 2021 Q4, I don’t see any mention of the SARS-CoV-2 pandemic and the role of non-pharmaceutical interventions in depressing the circulation of endemic respiratory pathogens or disruptions in seasonality (2021 appears to have a November peak compared to January peaks in previous years). This pattern has been seen in many places and should be added to the discussion.
Authors’ response: Thanks for the comment. We added lines 363-370
Lines 222 – 224: This would be a good place to include any changes you saw in the 1-4 age group during Q4 2021 – after the missed 2020-21 RSV season, thus delaying age of first infection/hospitalization.
Authors’ response: Thanks for the comment. We added lines 370-377
Line 230 – 238: I agree that improved recognition of the burden of RSV in older adults probably explains the increasing trend. But any thoughts on why your estimates for this age group are so much lower than the other studies?
Authors’ response: Thanks for the comment. We added lines 407-409
Lines 241 – 244: The authors do not mention the shift in cause of death from bronchiolitis to pneumonia after 2017 (which seems to also coincide with a shift in age). Would be worth mentioning in the results if it is going to be a focus in the discussion.
Authors’ response: It is mentioned in results, lines 130-132 “The chronological distribution of deaths indicates that no death occurred in infants hospitalized for RSV-related codes after the 2014-2015 season and that the 8 deaths registered from season 2016-2017 occurred exclusively in adults” and later “14 deaths were related to a pneumonia diagnosis: 1 occurred in the <1 year age-group, 1 in the 10-29 years age-group (the subject was 27 years old), 3 in the 50-69 years age-group, and 9 in the 70 years and older age-group.”
Lines 246 – 251: These sentences need revision so that punctuation is consistent. The number of deaths in 70+ is missing. It would also be more appropriate to give these detailed level figures in the results and focus on patterns in the discussion.
Authors’ response: Punctuation revised. Figure 3 added
Lines 251-256: I think this is the most important part of the paper. RSV is only recently being acknowledged as a serious threat in older adults. Highlighting this is extremely important, especially since vaccines for seniors will be available soon.
Reviewer 2 Report
brief summary
The study aims to analyze temporal trends and characteristics of hospitalization related to RSV in the Veneto region in Italy in the period from January 2007 to December 2021, based on hospital discharge records (HDRs) of public and accredited private hospitals. The strength of the paper is that it gives an overview of trends and characteristics of hospitalization related to RSV over 14 complete winter seasons. This study period is largely before, but also during the COVID-19 pandemic. It confirms high rates of hospitalization in infants and sheds light on the burden in the 70+ age-group in which they observe a considerable number of deaths.
General concept comments
This is a well-written manuscript that contains data from a large pre-COVID-19 pandemic, and the first year during the COVID-19 pandemic. Since the manuscript is submitted for review beginning of 2023, it would be an important added value (if not necessary) to include more recent data, in order to gain insight in the post COVID-19 situation. Especially after consulting Figure 2 with hospitalizations reaching the highest number of the entire series (p5 line 158).
The manuscript refers only to papers before march 2022, since then a lot of new manuscripts have been published, describing the RSV burden in hospitalized patients, please incorporate these recent publications in the introduction and discussion section, especially the papers investigation RSV burden in the elderly should be discussed and compared to the results that are presented in this study. Below you can find an non-exhaustive list of recently published manuscripts.
1. Sitthikarnkha P, Uppala R, Niamsanit S, Sutra S, Thepsuthammarat K, Techasatian L, Niyomkarn W, Teeratakulpisarn J. Burden of Respiratory Syncytial Virus Related Acute Lower Respiratory Tract Infection in Hospitalized Thai Children: A 6-Year National Data Analysis. Children (Basel). 2022 Dec 17;9(12):1990. doi: 10.3390/children9121990. PMID: 36553433; PMCID: PMC9776945.
2. Heppe-Montero M, Gil-Prieto R, Del Diego Salas J, Hernández-Barrera V, Gil-de-Miguel Á. Impact of Respiratory Syncytial Virus and Influenza Virus Infection in the Adult Population in Spain between 2012 and 2020. Int J Environ Res Public Health. 2022 Nov 9;19(22):14680. doi: 10.3390/ijerph192214680. PMID: 36429399; PMCID: PMC9690810.
3. Juhn YJ, Wi CI, Takahashi PY, Ryu E, King KS, Hickman JA, Yao JD, Binnicker MJ, Natoli TL, Evans TK, Sampathkumar P, Patten C, Luyts D, Pirçon JY, Damaso S, Pignolo RJ. Incidence of Respiratory Syncytial Virus Infection in Older Adults Before and During the COVID-19 Pandemic. JAMA Netw Open. 2023 Jan 3;6(1):e2250634. doi: 10.1001/jamanetworkopen.2022.50634. PMID: 36662530.
4. Suh M, Movva N, Jiang X, Bylsma LC, Reichert H, Fryzek JP, Nelson CB. Respiratory Syncytial Virus Is the Leading Cause of United States Infant Hospitalizations, 2009-2019: A Study of the National (Nationwide) Inpatient Sample. J Infect Dis. 2022 Aug 15;226(Suppl 2):S154-S163. doi: 10.1093/infdis/jiac120. PMID: 35968878; PMCID: PMC9377046.
5. Li Y, Wang X, Blau DM, Caballero MT, Feikin DR, Gill CJ, Madhi SA, Omer SB, Simões EAF, Campbell H, Pariente AB, Bardach D, Bassat Q, Casalegno JS, Chakhunashvili G, Crawford N, Danilenko D, Do LAH, Echavarria M, Gentile A, Gordon A, Heikkinen T, Huang QS, Jullien S, Krishnan A, Lopez EL, Markić J, Mira-Iglesias A, Moore HC, Moyes J, Mwananyanda L, Nokes DJ, Noordeen F, Obodai E, Palani N, Romero C, Salimi V, Satav A, Seo E, Shchomak Z, Singleton R, Stolyarov K, Stoszek SK, von Gottberg A, Wurzel D, Yoshida LM, Yung CF, Zar HJ; Respiratory Virus Global Epidemiology Network; Nair H; RESCEU investigators. Global, regional, and national disease burden estimates of acute lower respiratory infections due to respiratory syncytial virus in children younger than 5 years in 2019: a systematic analysis. Lancet. 2022 May 28;399(10340):2047-2064. doi: 10.1016/S0140-6736(22)00478-0. Epub 2022 May 19. PMID: 35598608; PMCID: PMC7613574.
The authors describe that analysis was performed on all the hospital discharge records of public and accredited private hospitals. Hospital discharge records were considered if included at least one of the listed ICD9-CM codes. Annual hospitalization rates (see also figure 4 were estimated by dividing the annual number of hospitalizations by the population of Veneto residents in the designated year as per data from Veneto Regional Authority of Statistics office. Please investigate and describe that all of the Hospitals in the Veneto region were using ICD-9 codes and not ICD-10 codes. If hospitals shifted from ICD9-CM to ICD-10 during the studied period, hospitalization rates can be biased.
Specific comments
1)p3 figure 1: The authors excluded 323 (4,4%) repeated admissions (two hospitalizations due to RSV for the same subject within a 30 day). This is indeed correct, on the other hand, a separate analysis on the variables impacting repeated admission (age, diagnosis code on admission,…) would give additional information on the burden of RSV.
2)p6 figure 3: The time of the lockdown is indicated in figure 3, please indicate at what time the other precautions (mask wearing, home working, contact restrictions…) were discontinued in this region.
3)For figure 3 and 4: please update with information from (at least) the complete 2021/2022 respiratory season.
4) p7 line 211: The authors describe that the positive trend in hospitalization rates in infants was only observed in Italy and argue that a possible explanation of this trend could be a parallel progressive increase in the number of tests requested by clinicians over the course of the study period as well as of improvements in laboratory techniques. If this is indeed the reason of the increasing trend of hospital admission rate in children, this trend should also be visible in other countries or please describe why this is only the case in the studied region. Please discuss these results using the more recent publications.
Author Response
Reviewer #2
brief summary
The study aims to analyze temporal trends and characteristics of hospitalization related to RSV in the Veneto region in Italy in the period from January 2007 to December 2021, based on hospital discharge records (HDRs) of public and accredited private hospitals. The strength of the paper is that it gives an overview of trends and characteristics of hospitalization related to RSV over 14 complete winter seasons. This study period is largely before, but also during the COVID-19 pandemic. It confirms high rates of hospitalization in infants and sheds light on the burden in the 70+ age-group in which they observe a considerable number of deaths.
General concept comments
This is a well-written manuscript that contains data from a large pre-COVID-19 pandemic, and the first year during the COVID-19 pandemic. Since the manuscript is submitted for review beginning of 2023, it would be an important added value (if not necessary) to include more recent data, in order to gain insight in the post COVID-19 situation. Especially after consulting Figure 2 with hospitalizations reaching the highest number of the entire series (p5 line 158).
Authors’ response: Thanks for the comment, but unfortunately data 2022-2023 is not also available
The manuscript refers only to papers before march 2022, since then a lot of new manuscripts have been published, describing the RSV burden in hospitalized patients, please incorporate these recent publications in the introduction and discussion section, especially the papers investigation RSV burden in the elderly should be discussed and compared to the results that are presented in this study. Below you can find an non-exhaustive list of recently published manuscripts.
- Sitthikarnkha P, Uppala R, Niamsanit S, Sutra S, Thepsuthammarat K, Techasatian L, Niyomkarn W, Teeratakulpisarn J. Burden of Respiratory Syncytial Virus Related Acute Lower Respiratory Tract Infection in Hospitalized Thai Children: A 6-Year National Data Analysis. Children (Basel). 2022 Dec 17;9(12):1990. doi: 10.3390/children9121990. PMID: 36553433; PMCID: PMC9776945.
Authors’ response: Included in introduction
- Heppe-Montero M, Gil-Prieto R, Del Diego Salas J, Hernández-Barrera V, Gil-de-Miguel Á. Impact of Respiratory Syncytial Virus and Influenza Virus Infection in the Adult Population in Spain between 2012 and 2020. Int J Environ Res Public Health. 2022 Nov 9;19(22):14680. doi: 10.3390/ijerph192214680. PMID: 36429399; PMCID: PMC9690810.
Authors’ response: Included in discussion
- Juhn YJ, Wi CI, Takahashi PY, Ryu E, King KS, Hickman JA, Yao JD, Binnicker MJ, Natoli TL, Evans TK, Sampathkumar P, Patten C, Luyts D, Pirçon JY, Damaso S, Pignolo RJ. Incidence of Respiratory Syncytial Virus Infection in Older Adults Before and During the COVID-19 Pandemic. JAMA Netw Open. 2023 Jan 3;6(1):e2250634. doi: 10.1001/jamanetworkopen.2022.50634. PMID: 36662530.
Authors’ response: Not included because this study is not about hospitalization.
- Suh M, Movva N, Jiang X, Bylsma LC, Reichert H, Fryzek JP, Nelson CB. Respiratory Syncytial Virus Is the Leading Cause of United States Infant Hospitalizations, 2009-2019: A Study of the National (Nationwide) Inpatient Sample. J Infect Dis. 2022 Aug 15;226(Suppl 2):S154-S163. doi: 10.1093/infdis/jiac120. PMID: 35968878; PMCID: PMC9377046.
Authors’ response: Included in discussion but we did not modify anything else in the article because results are similar to those of article already included.
- Li Y, Wang X, Blau DM, Caballero MT, Feikin DR, Gill CJ, Madhi SA, Omer SB, Simões EAF, Campbell H, Pariente AB, Bardach D, Bassat Q, Casalegno JS, Chakhunashvili G, Crawford N, Danilenko D, Do LAH, Echavarria M, Gentile A, Gordon A, Heikkinen T, Huang QS, Jullien S, Krishnan A, Lopez EL, Markić J, Mira-Iglesias A, Moore HC, Moyes J, Mwananyanda L, Nokes DJ, Noordeen F, Obodai E, Palani N, Romero C, Salimi V, Satav A, Seo E, Shchomak Z, Singleton R, Stolyarov K, Stoszek SK, von Gottberg A, Wurzel D, Yoshida LM, Yung CF, Zar HJ; Respiratory Virus Global Epidemiology Network; Nair H; RESCEU investigators. Global, regional, and national disease burden estimates of acute lower respiratory infections due to respiratory syncytial virus in children younger than 5 years in 2019: a systematic analysis. Lancet. 2022 May 28;399(10340):2047-2064. doi: 10.1016/S0140-6736(22)00478-0. Epub 2022 May 19. PMID: 35598608; PMCID: PMC7613574.
Authors’ response: Not included because not in scope.
The authors describe that analysis was performed on all the hospital discharge records of public and accredited private hospitals. Hospital discharge records were considered if included at least one of the listed ICD9-CM codes. Annual hospitalization rates (see also figure 4 were estimated by dividing the annual number of hospitalizations by the population of Veneto residents in the designated year as per data from Veneto Regional Authority of Statistics office. Please investigate and describe that all of the Hospitals in the Veneto region were using ICD-9 codes and not ICD-10 codes. If hospitals shifted from ICD9-CM to ICD-10 during the studied period, hospitalization rates can be biased.
Authors’ response: Thanks for the comment, all hospitals in the Veneto region use the ICD-9 codes and not the ICD-10 codes
Specific comments
1)p3 figure 1: The authors excluded 323 (4,4%) repeated admissions (two hospitalizations due to RSV for the same subject within a 30 day). This is indeed correct, on the other hand, a separate analysis on the variables impacting repeated admission (age, diagnosis code on admission,…) would give additional information on the burden of RSV.
Authors’ response: Thanks for the comment, we added the additional information in result section
2)p6 figure 3: The time of the lockdown is indicated in figure 3, please indicate at what time the other precautions (mask wearing, home working, contact restrictions…) were discontinued in this region.
Authors’ response: Thanks for the comment, we added 231-238
3)For figure 3 and 4: please update with information from (at least) the complete 2021/2022 respiratory season.
Authors’ response: Data 2022-2023 are not available
4) p7 line 211: The authors describe that the positive trend in hospitalization rates in infants was only observed in Italy and argue that a possible explanation of this trend could be a parallel progressive increase in the number of tests requested by clinicians over the course of the study period as well as of improvements in laboratory techniques. If this is indeed the reason of the increasing trend of hospital admission rate in children, this trend should also be visible in other countries or please describe why this is only the case in the studied region. Please discuss these results using the more recent publications.
Authors’ response: We believe that the increased trend in hospitalization rates is an effect of increased usage of tests and this is the opinion also of other authors. We are doubtful that, if it happens in Italy, it should be observed also in other countries. The reason why it has been observed in Italy has been provided but to speculate about why it has been observed only in Italy would be just speculating and it would not add value to the article.
Round 2
Reviewer 2 Report
Thank you for addressing the previous comments. Please find below some additional comments on the revised version of the manuscript:
1. Please read and correct the manuscript again, there are still many typo’s, e.g.:
Line 126: replace rispectively by respectively.
Line 137: replace “age groups 01-04 years” by “age groups 1-4 years”
Line 143: replace “the length increses with age” by “the length increases with age”
2. Line 168: “and 13 (56.5%) in subjects aged 70 years or more (4 for acute bronchiolitis and 9 for pneumonia).” Please also specify if these were primary or secondary diagnosis.
3. Line 146-181: please rework this part and include in the discussion the age group and time of the 5 deaths with primary diagnosis of acute bronchiolitis /pneumonia.
4. Figure 3: also indicate the death when patient had a primary diagnosis of pneumonia or acute bronchiolitis.
5. The added value of Figure 5 is low, the period of lockdown can also be depicted in figure 4, therefore I would delete figure 5.
6. Please describe how you define “season” (e.g. 2014-2015 season line 171 and lines 297-313)
7. Figure 2 was added to the manuscript and this figure is briefly described in the results section, but the discussion of these data is missing (lines 417-440).
Author Response
Thank you for addressing the previous comments. All the new comments are done on the revised version of the manuscript:
- Please read and correct the manuscript again, there are still many typo’s, e.g.:
Line 126: replace rispectively by respectively.
Line 137: replace “age groups 01-04 years” by “age groups 1-4 years”
Line 143: replace “the length increses with age” by “the length increases with age”
Done
- Line 168: “and 13 (56.5%) in subjects aged 70 years or more (4 for acute bronchiolitis and 9 for pneumonia).” Please also specify if these were primary or secondary diagnosis.
Done
- Line 146-181: please rework this part and include in the discussion the age group and time of
Done
- Figure 3: also indicate the death when patient had a primary diagnosis of pneumonia or acute bronchiolitis.
Done
- The added value of Figure 5 is low, the period of lockdown can also be depicted in figure 4, therefore I would delete figure 5.
Done
- Please describe how you define “season” (e.g. 2014-2015 season line 171 and lines 297-313)
done
- Figure 2 was added to the manuscript and this figure is briefly described in the results section, but the discussion of these data is missing (lines 417-440).
Done